# Design Optimization of Additive Manufactured Edgeless Simple Cubic Lattice Structures under Compression

**DOI:** 10.3390/ma16072870

**Published:** 2023-04-04

**Authors:** Kwang-Min Park, Young-Sook Roh

**Affiliations:** 1Construction Technology Research Centre, Construction Division, Korea Conformity Laboratories, Seoul 08503, Republic of Korea; kmpark@kcl.re.kr; 2Architectural Engineering Program, Department of Architectural Engineering, Seoul National University of Science and Technology, Seoul 01811, Republic of Korea

**Keywords:** additive manufacturing, simple cubic, edgeless lattice structure, fillet edgeless, multipipe edgeless, structural optimization, finite element analysis

## Abstract

This study proposed an optimization framework and methodologies to design edgeless lattice structures featuring fillet and multipipe functions. Conventional lattice structures typically experience stress concentration at the sharp edges of strut joints, resulting in reduced mechanical performance and premature failure. The proposed approach aimed to improve the compression behavior of lattice structures by introducing edgeless features. Through finite element analysis, the optimized fillet edgeless simple cubic unit cell with a fillet radius to strut radius ratio of 0.753 showed a 12.1% improvement in yield stress and a 144% reduction in stress concentration. To validate the finite element analysis, experimental compressive tests were conducted, confirming that the introduction of edgeless functions improved the compressive strength of lattice structures manufactured through additive manufacturing. The optimized fillet edgeless simple cubic lattice structure exhibited the most effective improvement. This approach has promising potential for lattice structure applications.

## 1. Introduction

Lattice structures received significant attention due to their attractive properties, including high specific strength and stiffness, ultra-low weight, excellent thermal management capabilities, excellent noise reduction capability, and their ability to absorb external impact forces through their internal structures [1,2,3]. The attractive properties of lattice structures fulfill the needs of manufacturing industry and provide unprecedented opportunities for structures with better manufacturing performance [4]. The properties of lattice structures determine their wide range of application fields. Lattice structures are often used in the structural design of aircraft, rocket, and other aerospace fields [5,6,7], as well as automotive fields [8,9,10]. Additionally, lattice structures possess porosity and integration potential with bone structures, enabling them to be designed in the shape of human tissue and bone to replace diseased organs [11,12,13].

Lattice structures consist of unit cells that are periodically arranged in a three-dimensional space [14,15,16]. These unit cells are defined by the dimensions and connectivity of their strut elements, which are connected at specific nodes [17]. The mechanical properties of lattice structures can be predicted based on their major design parameters, such as unit cell topology, relative density, and cell arrangement [18,19]. The unit cell is the smallest element that characterizes the entire lattice structure, and the mechanical properties of lattice structures depend primarily on the architecture of their unit cells when the type of base material and relative density are fixed [18].

The development of additive manufacturing (AM) technologies enabled the fabrication of lattice structures with complicated and novel architectures [20,21]. AM processes allow for the layer-by-layer construction of materials using a computer-aided design (CAD) model, enabling the creation of complex geometries that are not achievable with conventional manufacturing methods. The unique aspect of AM is its capability to fabricate hollow shapes with lattice structures. The AM processes are classified into seven categories, according to the ASTM F 2792-12 standard [22]: binder jetting (BJ), direct energy deposition (DED), material extrusion (ME), material jetting (MJ), powder bed fusion (PBF), sheet lamination (SL), and vat photopolymerization (VP).

Recent developments in AM processes for fabricating lattice structures involved various processing methods such as BJ, ME, MJ, PBF, and VP, among others. Of these methods, PBF in particular saw a growing number of applications for lattice structures due to its high precision and accuracy [4]. The PBF process involves creating a 3D digital model, which is divided into multiple layers, and these layers are successively bonded to form the final structure. During the process, a layer of powder material is deposited onto the build platform, and a heat source, such as a laser or electron beam, is used to fuse the first layer of the model according to the pre-designed 3D digital model. The build platform is then lowered and a new layer of powder is spread over the previous layer using a roller. This process is repeated until the entire model is constructed. Representative PBF methods are selective laser sintering (SLS) [23], selective laser melting (SLM) [24], and electron beam melting (EBM) [25]. PBF methods can be applied to a wide range of metallic materials, including Ti alloys, Al alloys, Ni alloys, Mg alloys, and steels [26,27,28,29].

Despite the superior properties of lattice structures, conventional designs with sharp-edged strut joints may suffer from stress concentration, which can lead to early yield or premature failure and a significant reduction in their mechanical performance [30,31]. Several studies proposed lattice structures with reduced stress concentration at the sharp edges of strut joints. For example, Bai et al. [29] proposed a new graded-strut body-centered cubic lattice structure with increased corner radius at the body-centered cubic nodes, which increased the plastic failure strength by at least 34.12% compared to the conventional body-centered cubic design. Li et al. [32] introduced a variable radius to generate a new diamond lattice structure, and the optimized radius improved the compressive modulus and compressive strength. Nazir et al. [33] developed a filleted kelvin lattice structure that improved energy absorption by 20% compared to those without fillets. Latture et al. [34] enhanced the mechanical properties of an octet truss lattice structure using nodal rounded fillets. Zhao et al. [35] improved the mechanical properties and energy absorption capacity of BCC lattice structures through structural optimization of minimal surfaces. However, these studies did not consider the relative density, which fluctuates when introducing fillet functions to the strut joints in the lattice structure.

Therefore, the research objectives of this study are to develop methodologies and frameworks that enable the optimization of edgeless lattice structure designs to address the aforementioned issues (Figure 1). Various edgeless lattice structures were designed using fillet and multipipe functions, and the effect of these structures under compression of constant relative density was analyzed. An optimized edgeless lattice structure topology was proposed. The details of the design methods for fillet and multipipe are introduced in Section 2.2 and Section 2.2.

## 2. Methodologies

### 2.1. Conventional Lattice Configuration

The simple cubic unit cell, which is one of the most representative types and superior under compression conditions, was investigated in this study [18]. The simple cubic cell was designed from node 1 to node 8, and it was designed with strut elements connected to specific nodes. In this study, a lattice-structured cube with nominal dimensions of 20 mm × 20 mm × 20 mm was designed (Figure 2). The length of the unit cell cube edge was denoted as s. The unit cell strut was created in a circular section to generate a volume in the unit cell. The specific relative density was determined by controlling the strut radius (r). By setting the ratio of strut radius to edge length (r/s) as a variable, it was possible to control the relative densities of the unit cells (Figure 3). The length of the unit cell cube edge (s) was designed to be 20 mm in consideration of previous study [18] and case studies related to the strength of the lattice structure [36,37,38,39,40]. By using r/s as a design variable, it can be commonly applied when designing various types of unit cell morphologies and sizes, allowing for a common approach regardless of the specific morphology or dimensions of the unit cell [18].

To fabricate the lattice structure, the unit cells were arranged by repeating the lattice points, which is referred to as the pattern design. A lattice structure can be created from an array of repeated unit cells using direct patterning, in which the unit cells are directly generated by repeating the unit cells in three dimensions (along the *x*-, *y*-, and *z*-axes). For example, a 2 ea × 2 ea × 2 ea lattice structure can be constructed by repeating the unit cell twice in each coordinate axis (Figure 2), which is expressed as {2 × 2 × 2} ea hereafter.

To automate the process, we developed the KCL lattice structure generator, a plugin for Rhinoceros (Rhino 7, Robert McNeel & Associates, Seattle, WA, USA). The KCL lattice structure generator can generate geometries for various types of strut-based unit cell lattice structures. The graphical user interface consists of two main displays: (1) a lattice structure display and (2) a design parameter display (Figure 4a). The design parameters display consists of seven main tabs (Figure 4b): (1) unit cell topology, (2) boundary size, (3) unit cell arrangement, (4) strut radius, (5) unit center of {3 × 3 × 3} ea, (6) fillet edgeless lattice, and (7) multipipe edgeless lattice. This plugin has various unit cell topologies and can adjust the lattice structure boundary size, unit cell arrangement, and strut radius of the unit cell. It also includes a unit center of {3 × 3 × 3} ea, a fillet edgeless function, and a multipipe edgeless function to create different designs for edgeless lattice structures.

However, in the case of the body center unit cell (Figure 5a), diagonal struts at the edge of the boundary made it impossible to implement fillet or multipipe edgeless functions at the corners when applied to unit cells. Therefore, we arranged the unit cells in a {3 × 3 × 3} ea configuration and extracted the central unit cell for analysis (Figure 5b). We termed this process “the unit center of {3 × 3 × 3} ea”. By applying this process, we created an edgeless unit cell at the corner of the boundary (Figure 5c). The relative density of the optimized edgeless lattice structures could be controlled by calculating the relative density in the unit cell extracted from the proposed novel method.

### 2.2. Fillet Edgeless Lattice Configuration

The sharp edges of the strut joints of the simple cubic unit cells are on their interior corners. Therefore, the geometric function of the fillet is concave. A comparison of a conventional simple cubic unit cell and a fillet edgeless simple cubic unit cell is demonstrated in Figure 6, where s is the length of the unit cell cube edge, r is the strut radius, and $\overset{-}{r}$ is the fillet radius. The KCL lattice structure generator plugin for Rhinoceros used the rolling ball blends fillet method [41] from the common function in Rhinoceros, a widely used geometric modeling method for generating fillets in sharp corners.

The relative density increased when the fillet radius increased within the same strut radius; the relative density can be maintained constant by reducing the strut radius. In the fillet edgeless function, the unit cell was designed according to the relative densities by setting the strut and fillet radiuses as parameters. The fillet edgeless lattice structures with a constant relative density were designed based on the ratio of the strut radius to the edge length (r/s) and the ratio of the fillet radius to the edge length ($\overset{-}{r}/s$). Consequently, by utilizing the ratio of the fillet radius to the strut radius ($\overset{-}{r}/r$), it was possible to design an edgeless lattice structure with constant relative density. In this study, various morphologies of a fillet edgeless simple cubic unit cell with constant relative density were investigated, and a fillet edgeless simple cubic unit cell morphology with mechanical superiority under compression conditions was derived through finite element analysis (FEA) validation. Subsequently, a lattice structure was designed by arranging the optimized unit cells in a {3 × 3 × 3} ea pattern. The mechanical properties of the unit cell were used to represent the properties of the lattice structure and were utilized for the design and analysis of the entire structure [42,43,44,45]. The FEA validation method is described in Section 2.4.

Table 1 and Figure 7 show the database for the fillet edgeless simple cubic unit cell design with relative densities of 0.1, 0.2, and 0.3. In this study, $\overset{-}{r}/s$ was provided from 0.01 to 0.25, in increments of 0.01. The morphologies of 12 fillet edgeless simple cubic unit cells with a relative density of 0.2 and $\overset{-}{r}/s$ ranging from 0.02 to 0.24 with an increment of 0.02 are shown in Figure 8, which were selected for the FEA validation. In this study, the length of the unit cell cube edge (s) was fixed at 20 mm.

### 2.3. Multipipe Edgeless Lattice Configuration

The multipipe function in Rhinoceros created a subdivided pipe frame with smooth junctions from intersecting curves. The input parameters are shown in Figure 9a. In this study, the node size, end offset, and strut size were reviewed as variable parameters, and the definitions of each variable parameter on the morphology control of the unit cell are shown in Figure 9b. The node size represents the radius of each node point. The end offset is the distance of the first edge loop away from the node, expressed as a multiplier of the node size. The strut size is the radius of the struts, expressed as multiples of the node size. The relative density of the multipipe edgeless unit cell was controlled by setting the node size and end offset as the variable parameters. Here, the strut size was set equal to the node size. In this study, the unit center of {3 × 3 × 3} ea process was processed.

A comparison between a conventional simple cubic unit cell and a multipipe edgeless simple cubic unit cell is presented in Figure 10. In this figure, s is the length of the unit cell cube edge, r is the strut radius, nodesize is the radius of the node point, and endoffset is the distance from the node.

The relative density of the lattice structure increased as the nodesize increased within the same endoffset. To control constant relative densities, the endoffset was reduced. Multipipe edgeless lattice structures with a constant relative density were designed by the ratio of the nodesize to the edge length (nodesize/s). In this study, various morphologies of multipipe edgeless simple cubic unit cells with a constant relative density were investigated, and a morphology with superior mechanical properties under compression conditions was derived through validation using finite element analysis (FEA).

Figure 11 depicts the design database for multipipe edgeless simple cubic unit cells with a relative density ranging from 0.1 to 0.45. The nodesize/s was provided in increments of 0.001, ranging from 0.1630 to 0.1580. The smallest endoffset condition was used to implement the morphology, as well as the condition value at which the morphology change occurred due to the endoffset. Figure 12 shows the morphologies of 9 multipipe edgeless simple cubic unit cells with a relative density of 0.2. The length of the unit cell cube edges (s) was fixed at 20 mm.

### 2.4. Finite Element Analysis

Nonlinear static analysis using SolidWorks 2021 (Dassault Systems, Paris, France) was performed to determine the optimized edgeless lattice structures for axial compressive load. A fixed boundary condition was applied to the bottom surface of the structure, and a compression force was applied to the top surface in a direction orthogonal to it (Figure 13). The Fourier finite-element plus (FFEPLUS) iterative solver was used for the nonlinear static analysis. After mesh convergence analysis, solid mesh elements were applied to the lattice structures at a mesh element size of 0.2 mm. The maximum compression force was achieved at the point where the von Mises stress met the yield stress of the material (Figure 13). In this study, the yield stress was defined as the compression force achieved at the yield point divided by the upper area of the specimen, which was 400 mm^2^ (20 mm × 20 mm). All stress values were calculated based on the elemental mean values.

The material properties of the Ti alloy (Ti6Al4V) used in this study were derived experimentally: an elastic modulus of 119.0 GPa, yield strength of 1125.0 MPa, tensile strength of 1200.0 MPa, and Poisson’s ratio of 0.34 were achieved and used for numerical analysis. Therefore, the yield force was evaluated as the point at which the mesh element reached 1125.0 MPa. To evaluate the effect of edgeless lattice structures on stress concentration reduction, the number of meshes subjected to stress ranging from 1012.5 MPa to 1125.0 MPa, corresponding to the top 10% of the von Mises stress distribution, were analyzed [46]. This number is referred to as the “Number of meshes in the top 10% stress range”. The percentage of the number of meshes in the top 10% stress range to the total number of meshes is expressed as “Percentage of the number of meshes in the top 10% of the stress range, %”.

### 2.5. Specimens for Additive Manufacturing and Experimental Compressive Tests

The specimens were manufactured using SLM (Metal3D Metalsys 250E, Ulsan, Republic of Korea). In the manufacturing process, spherical Ti6Al4V Grade 5 titanium powder with particle sizes ranging from 10 μm to 45 μm was used. The layer thickness was 0.02 mm, and the following parameters were applied: laser power of 185 W, scanning speed of 1100 mm/s, and hatching distance of 90 µm. The building volume of the manufacturing equipment was 250 mm × 250 mm × 250 mm. The specimens were placed on the same layer of the building volume and manufactured in batches, and the manufacturing time was approximately 20 h.

The compression test was performed using a universal testing machine (UTM, Instron Universal Testing Machine, Norwood, MA, USA), with a maximum loading capacity of 100 kN. The force was applied at a compression speed of 0.01 mm/s in accordance with ISO 13314 [47]. The compressive strength was derived as the average of two specimens. The loading was applied until collapse, and the maximum compressive load at that point was determined. The loading area of the specimens for compressive strength evaluation was the upper loading area of the specimen. In this study, the compressive strength was defined as the maximum load divided by the upper loading area of the specimen, which was 400 mm^2^ (20 mm × 20 mm).

## 3. Results and Discussion

### 3.1. Fillet Edgeless Simple Cubic Unit Cell

#### 3.1.1. Yield Stress Improvement Effect with Fillet Edgeless Function

Figure 14 shows the von Mises yield stress distribution obtained through FEA, in which the compressive force was achieved at the point of various fillet edgeless simple cubic unit cells. In a conventional simple cubic structure, the von Mises yield stress is concentrated at the sharp edges of the strut joints. However, in the fillet edgeless simple cubic unit cell, as the fillet radius increases, the stress concentration is reduced, and the stress distribution widens throughout the unit cell structure. Table 2 and Figure 15 summarize the von Mises yield stress obtained through FEA and the stress concentration according to the fillet edgeless simple cubic unit cells by design parameters.

In the conventional simple cubic unit cell, the yield stress was 28.0 MPa (Figure 14a). In the fillet edgeless simple cubic unit cells, when the ratios of the fillet radius to the strut radius ($\overset{-}{r}/r$) were 0.1233 and 0.2469, the yield stress decreased to 27.7 MPa and 27.8 MPa, respectively (Figure 14b,c). This result indicates that when $\overset{-}{r}/r$ was smaller than 0.2469, the yield stress could not be improved by the fillet edgeless design. However, when $\overset{-}{r}/r$ was 0.3713 or higher, the yield stress was improved by the fillet edgeless design. The yield stress increased significantly when $\overset{-}{r}/r$ increased from 0.3713 to 0.7533, and the highest yield stress was 31.4 MPa at $\overset{-}{r}/r$ of 0.7533 (Figure 14g).

As a result, the yield stress of the fillet edgeless simple cubic unit cell with $\overset{-}{r}/r$ = 0.7533 was improved by 12.1% compared to the conventional simple cubic unit cell. However, when $\overset{-}{r}/r$ was higher than 0.7533, the yield stress gradually decreased as $\overset{-}{r}/r$ increased. When $\overset{-}{r}/r$ = 1.5937, the yield stress was 30.3 MPa. Therefore, fillet edgeless optimization was possible in the range of 0.753 ≤ $\overset{-}{r}/r$ ≤ 1.109, and a yield improvement of 12.1% could be achieved.

#### 3.1.2. Reducing Stress Concentration with the Edgeless Unit Cell Design

The total number of meshes of the conventional simple cubic unit cell was 521,249, and the number of meshes in the top 10% stress range was 5032, which accounted for 0.97% of the total meshes. As shown in Figure 14, the stress concentration was located at the sharp edges of the strut joints. When $\overset{-}{r}/r$ was 0.123 and 0.247, the percentage of the number of meses in the top 10% of the stress range was 0.95% and 0.98%, respectively. This result indicates that when $\overset{-}{r}/r$ was smaller than 0.247, the stress concentration was not reduced by the fillet edgeless function. However, when $\overset{-}{r}/r$ was higher than 0.3713, the percentage of the number of meshes in the top 10% of the stress range increased with the $\overset{-}{r}/r$. When $\overset{-}{r}/r$ increased from 0.3713 to 0.7533, the percentage of the number of meshes in the top 10% of the stress range increased significantly from 1.34% to 2.37%. The percentage of the number of meshes gradually increased when $\overset{-}{r}/r$ was higher than 0.7533. The results indicate that it was effective to reduce stress concentration by increasing the $\overset{-}{r}/r$ using fillet edgeless function.

Improving the edges to a smoother shape can reduce stress concentrations caused by joint edges [48,49]. As the fillet radius increased, the percentage of the number of meshes in the top 10% stress range also continued to increase. Ultimately, this means that the fillet edgeless design method is effective in reducing stress concentrations and enhancing stress. However, to increase the fillet radius, the strut radius must be reduced in order to maintain a constant relative density in the lattice structure.

### 3.2. Multipipe Edgeless Simple Cubic Unit Cell

#### 3.2.1. Yield Stress Improvement Effect with Multipipe Edgeless Function

Figure 16 shows the von Mises yield stress distribution obtained by FEA, in which the compressive force was achieved at the point of various multipipe edgeless simple cubic unit cells. Table 3 and Figure 17 summarize the von Mises yield stress and the stress concentration according to the multipipe edgeless simple cubic unit cells by design parameters. In the conventional simple cubic unit cell, the yield stress was 28.0 MPa (Figure 16a). For the multipipe edgeless simple cubic unit cells, when the ratio of the node size to the edge length (nodesize/s) was 0.1634 and 0.1630, the yield stress decreased to 26.1 MPa and 26.9 MPa, respectively (Figure 16b,c). This result indicates that when nodesize/s was smaller than 0.1630, the yield stress was not improved by the multipipe edgeless function. From nodesize/s of 0.1620 to 0.1580, the yield stress was improved, but the improvement was insignificant (improvement rate of 2.9%) (Figure 16f,g). Furthermore, when the nodesize/s was 0.1462 and 0.1019, the yield stress decreased to 27.8 MPa and 23.1 MPa, respectively, which was lower than the conventional simple cubic unit cell (Figure 16i,j). As a result, optimization by the multipipe edgeless function was possible in the range of 0.1610 ≤ nodesize/s ≤ 1.1580, and a yield stress improvement of 2.9% can be achieved.

In addition, the endoffset value was increased to maintain a constant relative density when the ratio of the node size to the edge length was increased. As a result, a two-stage morphology change was observed for the multipipe edgeless simple cubic unit cells while increasing the endoffset. However, due to the absence of an open-source algorithm for multipipe components, a detailed analysis is not presented in this study.

#### 3.2.2. Reducing Stress Concentration with Edgeless Unit Cell Function

The total number of meshes of the conventional simple cubic unit cell was 521,249, and the number of meshes in the top 10% stress range was 5032, which was 0.97% of the total mesh. As shown in Figure 17, when the nodesize/s was 0.1634 and 0.1630, the percentage of the number of meshes in the top 10% of the stress range was 0.68% and 0.90%, respectively. This result indicates that when nodesize/s was smaller than 0.1630, the stress concentration was not reduced by the multipipe edgeless function. However, when the nodesize/s increased from 0.1610 to 0.1580, the percentage of the number of meshes in the top 10% of the stress range increased significantly from 1.33% to 1.45%.

### 3.3. Optimized Edgeless Lattice Structure

The optimal edgeless unit cell type was determined through FEA analysis, as presented in Table 4. Using a relative density of 0.2 and dimensions of 20 mm × 20 mm × 20 mm, lattice structures were formed by arranging the optimized unit cells in a {3 × 3 × 3} ea array, to achieve the maximum compression force at the yield stress of the lattice structures [18]. Therefore, the edge length of the unit cells in the lattice structures was 1/3 of that of the optimal edgeless unit cell.

The yield stress of the conventional simple cubic unit cell, the optimized fillet edgeless simple cubic unit cell, and the optimized multipipe edgeless simple cubic unit cell at the relative density of 0.2 were 28.0 MPa, 31.4 MPa, and 28.8 MPa, respectively. In addition, the yield stress of the conventional simple cubic lattice structure, the optimized fillet edgeless simple cubic lattice structure, and the optimized multipipe edgeless simple cubic lattice structure at the relative density of 0.2 were 44.4 MPa, 48.8 MPa, and 44.4 MPa, respectively. This indicates that the yield stress of the lattice structures increased by 158.6%, 155.4%, and 154.2% compared to each unit cell, respectively.

The yield stress of the optimized fillet edgeless simple cubic unit cell was improved by 12.1% compared to the conventional simple cubic unit cell. In addition, the yield stress of the optimized fillet edgeless simple cubic lattice structure was improved by 9.9% compared to the conventional simple cubic lattice structure. However, the yield stress improvement of multipipe edgeless function was only 2.9% and 0.0% in unit cell and lattice structure, respectively.

### 3.4. Validation through Experimental Compressive Test

To validate the FEA results obtained in Section 3.3, specimens were manufactured using the additive manufacturing process described in Section 2.5, and compression tests were performed. Figure 18 shows the specimen’s shape manufactured using SLM additive manufacturing with Ti6Al4V material. Figure 19 depicts the deformation behavior of the lattice structures during the compression test.

In the case of the conventional simple cubic unit cell, initial fracture occurred at the node edge of the upper part of the lattice structure (Figure 18, yellow dotted circles prior to collapse), which corresponded to the stress concentration areas indicated by the FEA results (Table 4). As the compression load was increased, progressive collapse occurred starting from the location of initial fracture. On the other hand, for the optimized fillet edgeless simple cubic unit cell, an ideal failure mode of compression failure was observed. The initial fracture did not occur at the node edge but rather the central part of the vertical strut buckled and failed. Similarly, for the optimized multipipe edgeless simple cubic unit cell, the initial fracture occurred in proximity to the node edge.

As a result, it was confirmed that the introduction of edgeless functions affects the deformation and fracture mechanisms of the same simple cubic unit cell. It was validated that prior to collapse, initial fractures occurred at the stress concentration location derived through FEA. However, it is necessary to accumulate FEA and validation experimental tests for detailed deformation beyond the elastic region into the plastic region and final fracture.

Table 5 and Figure 20 present the results of experimental compressive strength tests. In the unit cell concept, the compressive strengths of the conventional simple cubic, fillet edgeless simple cubic, and multipipe edgeless simple cubic were measured as 78.7 MPa, 88.4 MPa, and 76.3 MPa, respectively. In the lattice structure concept, which consisted of {3 × 3 × 3} ea unit cell arrays, the compressive strengths were measured as 118.5 MPa, 131.0 MPa, and 117.8 MPa, respectively. These results are consistent with the FEA results, showing that the optimized edgeless structures had the most effective improvement in compressive strength. Additionally, the ratios of relative yield stress and relative compressive strength were measured at the same level based on the conventional simple cubic. The experimental compressive strength was measured as 2.65 to 2.81 times the yield stress from FEA. It was observed that there was a difference between the yield stress obtained through FEA and the maximum compressive strength of lattice structures obtained through experimental testing until fracture point. Therefore, there is a need to develop a methodology for evaluating the relationship between yield stress obtained through FEA and the maximum compressive strength obtained through the experimental testing of lattice structures.

### 3.5. Limitations of the Study

This study had certain limitations, which are outlined as follows. Firstly, the edgeless optimization methodology was only evaluated for the simple cubic unit cell. In order to fully comprehend the overall characteristics of lattice structures, it is essential to ensure diversity in the edgeless unit cell. Secondly, the FEA analysis was limited to the yield strength of the material in the elastic region, and further analysis is required to study the behavior of lattice structures in the plastic and fracture regions. Lastly, the compressive strength of the lattice structure was determined solely by the value at which it was destroyed. Therefore, conducting experiments to observe accurate stress–strain curves and to study the elastic, plastic, and fracture behavior of lattice structures is necessary.

## 4. Conclusions

This study developed methodologies and frameworks for the design optimization of edgeless lattice structures under compression by introducing fillet and multipipe functions. The investigation revealed that the edgeless unit cell topology plays a crucial role in determining the mechanical properties of the lattice structures with a constant relative density. Consequently, under the constant relative density conditions, it was confirmed that the optimized fillet edgeless simple cubic lattice structure was effective in improving yield stress, compressive strength, and reducing stress concentration. The main findings of this study are summarized as follows.

Methodologies and frameworks for design optimization were established to achieve optimized edgeless lattice structures assisted by the lattice structure generator plugin for Rhinoceros, which is called the KCL lattice structure generator. The results were achieved by: (i) introducing the derivation of the representative relative density of the edgeless unit cell by extracting the central unit cell from the {3 × 3 × 3} ea arranged lattice structure; (ii) introducing fillet edgeless function by design variable parameters (i.e., the ratio of the fillet radius to the strut radius ($\overset{-}{r}/r)$); (iii) introducing multipipe edgeless function by design variable parameters (i.e., the ratio of the node size to the edge length (nodesize/s) and endoffset); and (iv) introducing a new method for evaluating the degree of stress concentration by the number of meshes subjected to stress, which corresponds to the top 10% of the von Mises stress distribution.Through nonlinear static FEA, it was confirmed that the optimized fillet edgeless simple cubic unit cell improved the properties of yield stress and stress concentration. The yield stress of the fillet edgeless simple cubic unit cell with 0.753 ≤ $\overset{-}{r}/r$ ≤ 1.109 was improved by approximately 12.1% compared to the conventional simple cubic unit cell. When 0.753 ≤ $\overset{-}{r}/r$ ≤ 1.109, the percentage of the number of meshes in the top 10% of the stress range ranged from 2.37% to 2.52%, which means that it was effective in reducing stress concentration, compared to that of the conventional simple cubic at 0.97%.Through nonlinear static FEA, it was confirmed that the optimized multipipe edgeless simple cubic unit cell improved the properties of yield stress and stress concentration. In the optimized multipipe edgeless parameters presented in this study, the yield stress of multipipe edgeless simple cubic unit cell with 0.1610 ≤ nodesize/s ≤ 1.1580 was improved by approximately 2.9% compared to that of the conventional simple cubic unit cell. However, it was observed that the improvement was less effective than the fillet edgeless function.The experimental compressive tests confirmed that the introduction of edgeless functions to lattice structures improved their compressive strength. The optimized fillet edgeless simple cubic structure showed the most significant improvement compared to conventional simple cubic and multipipe edgeless structures. The results of the compressive strength tests were consistent with the FEA results, and initial fractures occurred at stress concentration locations derived from FEA. However, further FEA and experimental tests are needed to study the detailed deformation beyond the elastic region into the plastic region and final fracture.

This study produced three main results: (1) the proposal of an edgeless design methodology for lattice structures, (2) validation through FEA indicating an improvement in the yield stress and stress concentration of edgeless lattice structures, and (3) validation through experimental tests demonstrating an improvement in the maximum compressive strength of edgeless lattice structures. However, further investigation is required to expand the scope of FEA review to include fracture mode, and to observe the deformation of the lattice structure through detailed stress–strain curve analysis in experimental tests. After examining the above, the compression behavior in the elastic and plastic ranges of the lattice structure should be observed. Additionally, this study was limited to the simple cubic unit cell topology, and further review is required for various unit cell topologies.

## Figures and Tables

**Figure 1 materials-16-02870-f001:**
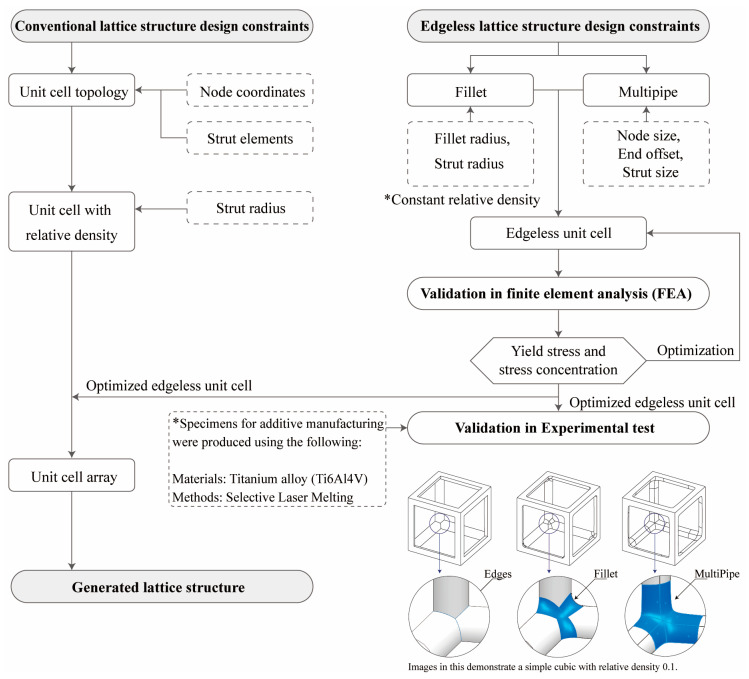
Optimization method of the edgeless lattice structure configuration and the scheme of the study.

**Figure 2 materials-16-02870-f002:**
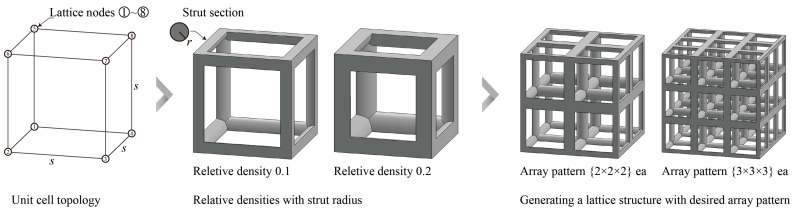
Scheme of the conventional simple cubic lattice structure configuration method.

**Figure 3 materials-16-02870-f003:**
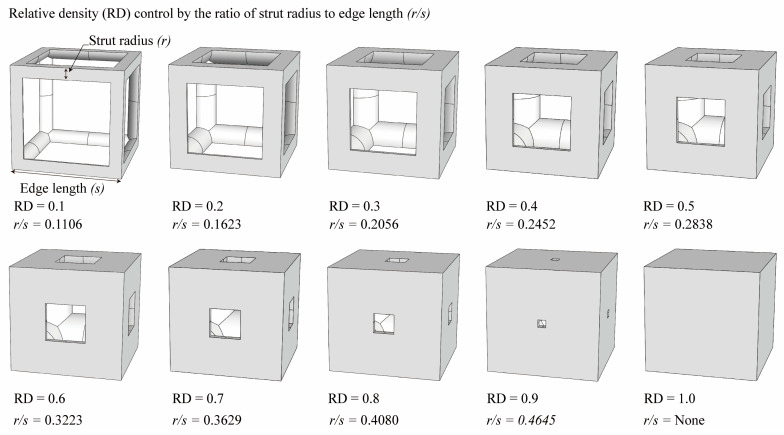
Relative density (RD) of the simple cubic unit cell controlled by the ratio of strut radius to edge length.

**Figure 4 materials-16-02870-f004:**
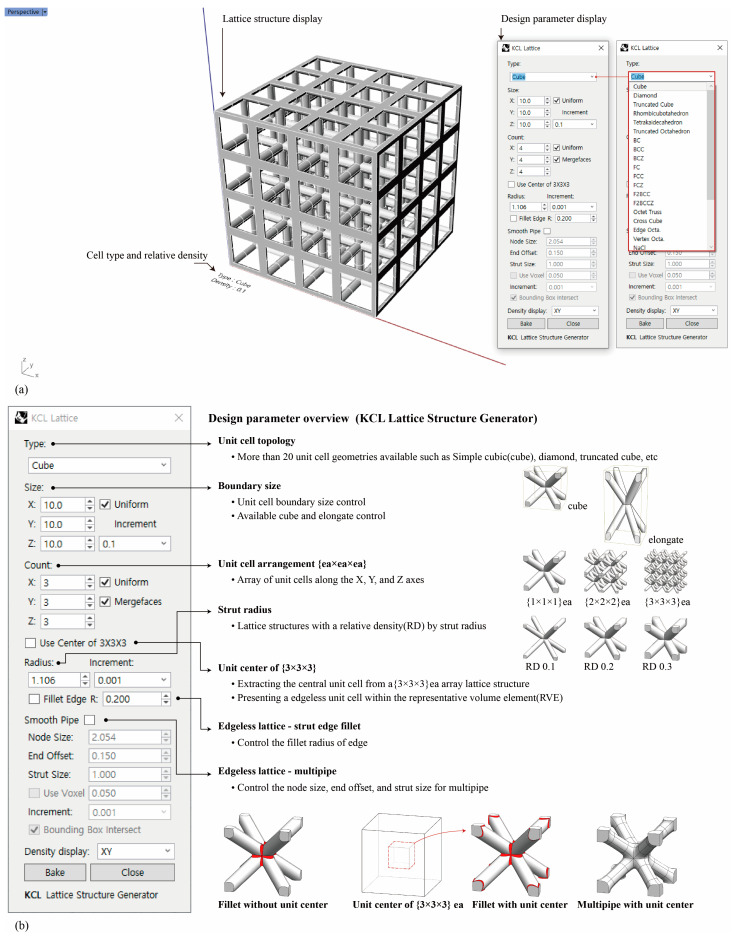
Screenshots of the KCL lattice structure generator plugin for Rhinoceros: (**a**) graphical user interface; (**b**) design parameters.

**Figure 5 materials-16-02870-f005:**
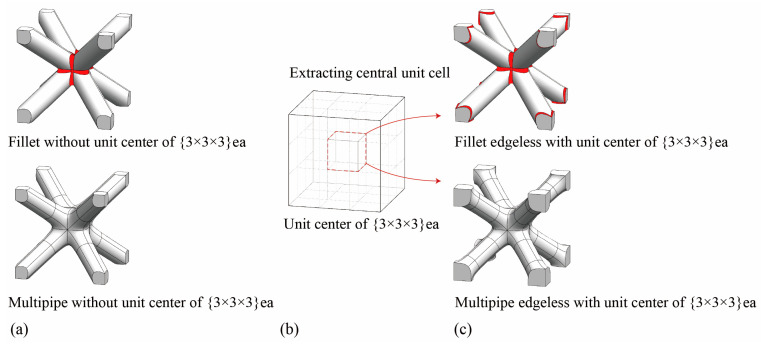
Concept of the unit center of {3 × 3 × 3} ea process: (**a**) fillet and multipipe edgeless functions applied to a unit cell; (**b**) the unit center of {3 × 3 × 3} ea process; (**c**) fillet and multipipe edgeless unit cells created using the unit center of {3 × 3 × 3} ea process. Red-colored areas indicate where the edgeless functions were implemented.

**Figure 6 materials-16-02870-f006:**
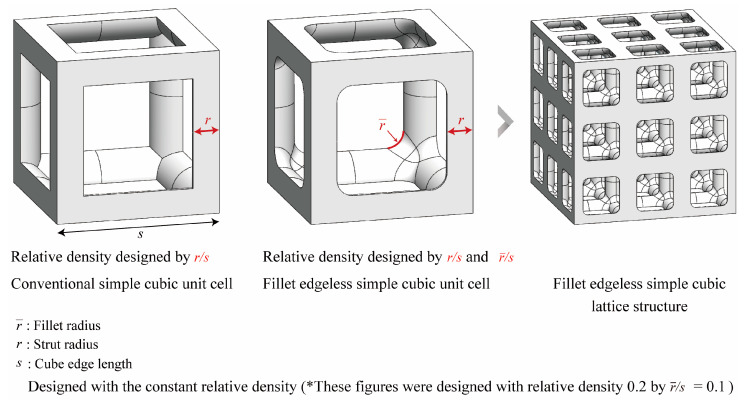
Schematics of the fillet edgeless simple cubic lattice structure. The structures were designed with a constant relative density of 0.2 and $\overset{-}{r}/s$ = 0.1.

**Figure 7 materials-16-02870-f007:**
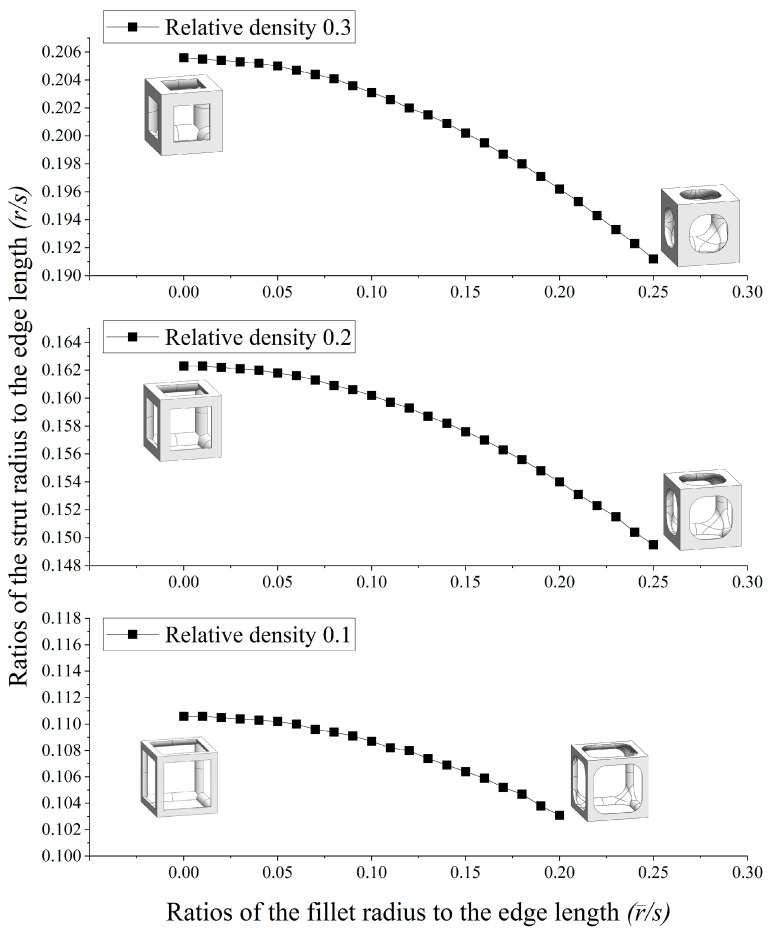
Relationship between the ratio of the strut radius to the edge length and the ratio of the fillet radius to the edge length at constant relative densities for fillet edgeless simple cubic unit cells.

**Figure 8 materials-16-02870-f008:**
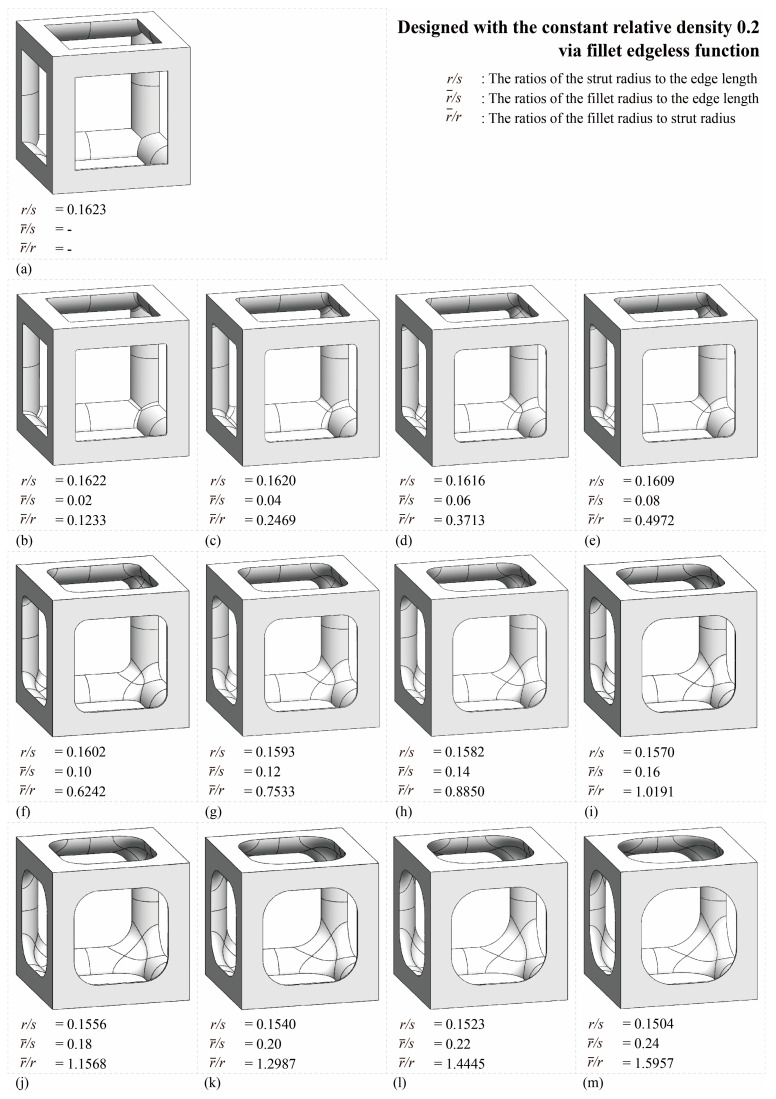
Various morphologies of fillet edgeless simple cubic unit cells with a relative density of 0.2: (**a**) conventional simple cubic unit cell; (**b**–**m**) fillet edgeless simple cubic unit cells.

**Figure 9 materials-16-02870-f009:**
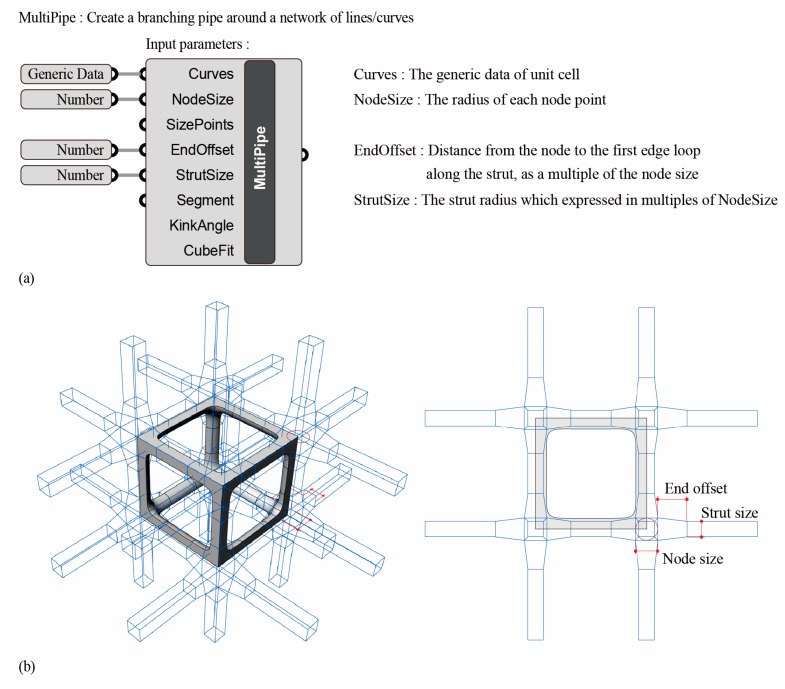
Multipipe function in Rhinoceros: (**a**) input parameters; (**b**) position of each variable parameter on the morphology control of the unit cell.

**Figure 10 materials-16-02870-f010:**
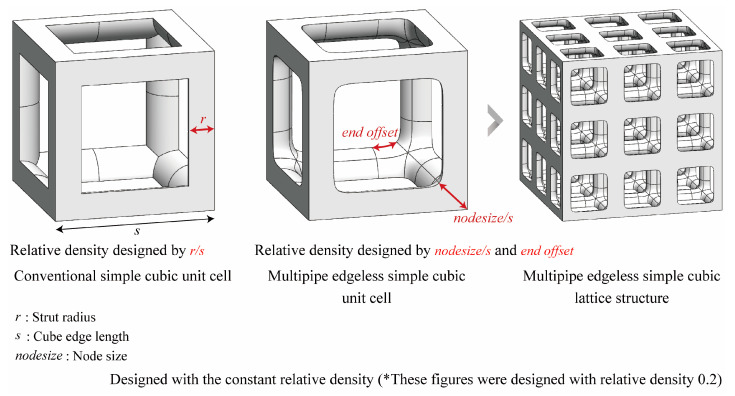
Schematics of the multipipe edgeless simple cubic lattice structure. The structures were designed with a constant relative density of 0.2.

**Figure 11 materials-16-02870-f011:**
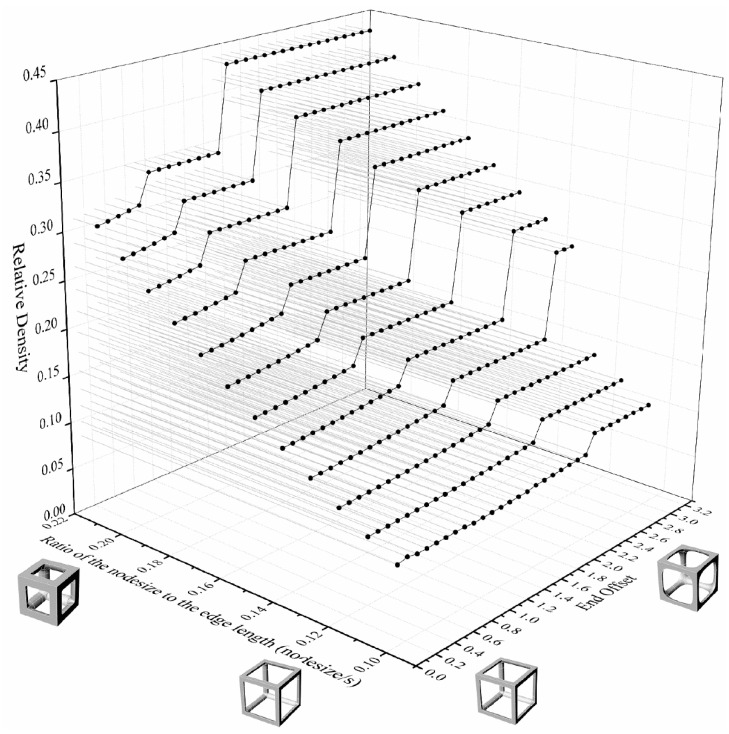
Design database for multipipe edgeless simple cubic lattice structure with the ratios of the node size to the edge length and end offset.

**Figure 12 materials-16-02870-f012:**
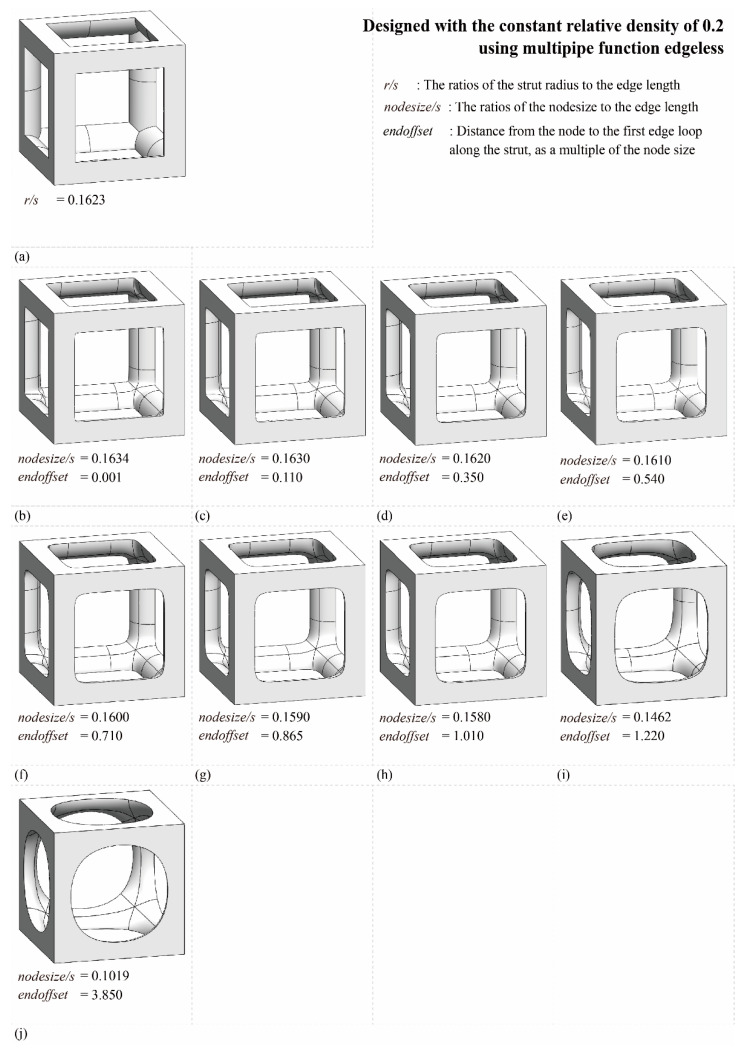
Various morphologies of multipipe edgeless simple cubic unit cells with a relative density of 0.2: (**a**) conventional simple cubic unit cell; (**b**–**j**) multipipe edgeless simple cubic unit cells.

**Figure 13 materials-16-02870-f013:**
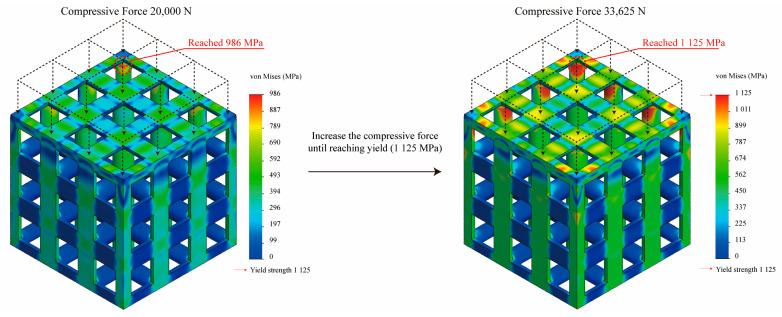
Yield stress derivation using finite element analysis.

**Figure 14 materials-16-02870-f014:**
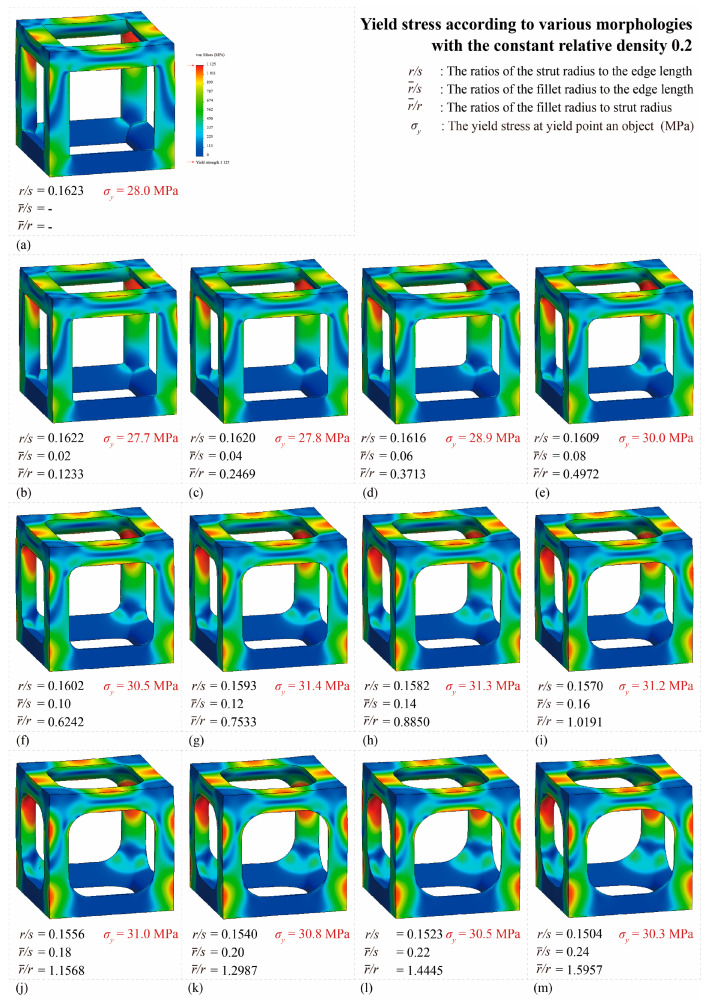
Yield stress according to various fillet edgeless simple cubic unit cells morphologies with a constant relative density of 0.2: (**a**) conventional simple cubic unit cell; (**b**–**m**) fillet edgeless simple cubic unit cells.

**Figure 15 materials-16-02870-f015:**
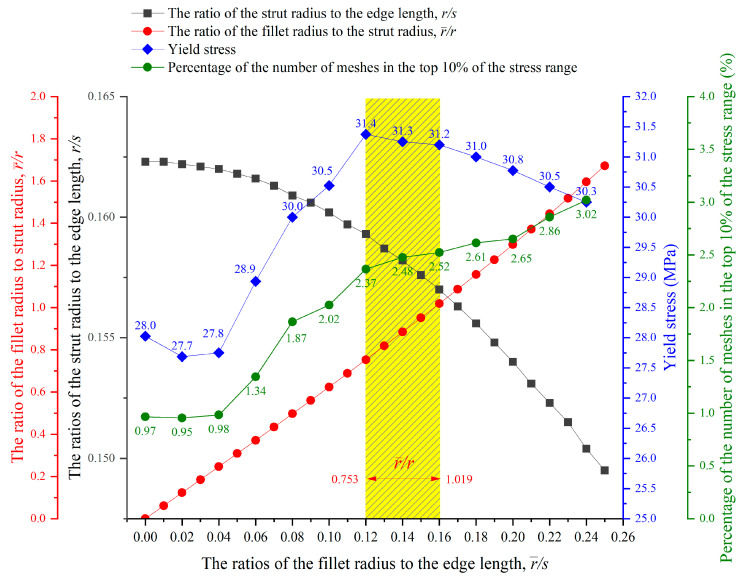
FEA results for the fillet edgeless simple cubic unit cells.

**Figure 16 materials-16-02870-f016:**
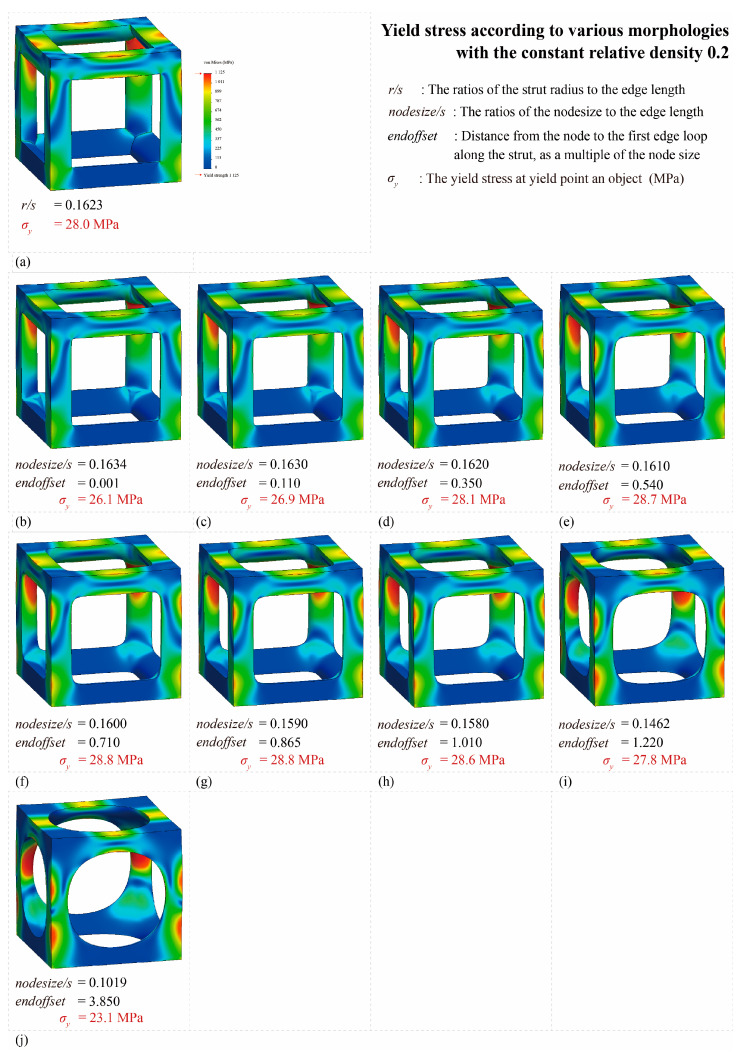
Yield stress according to various multipipe edgeless simple cubic unit cells morphologies with a constant relative density of 0.2: (**a**) conventional simple cubic unit cell; (**b**–**j**) multipipe edgeless simple cubic unit cells.

**Figure 17 materials-16-02870-f017:**
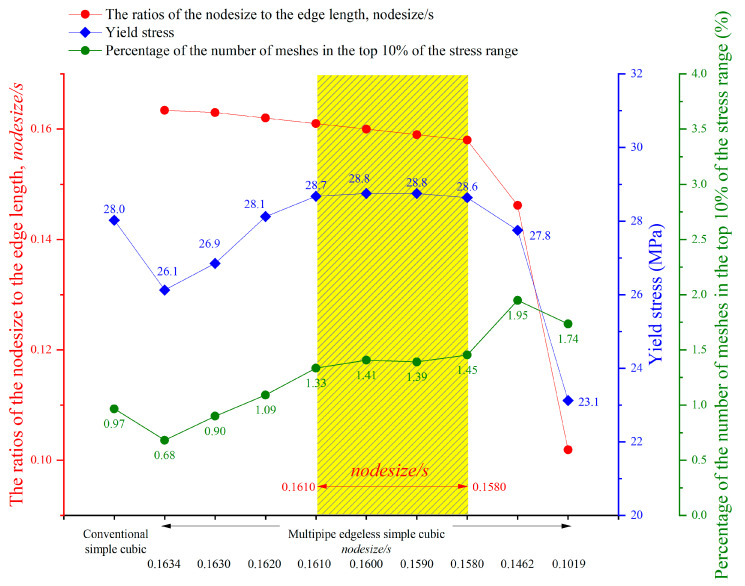
FEA results for the multipipe edgeless simple cubic unit cells.

**Figure 18 materials-16-02870-f018:**
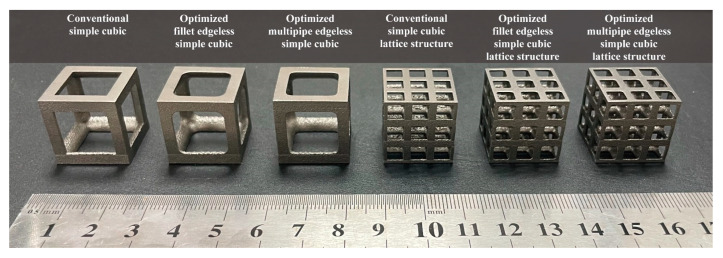
The additive manufactured specimens with optimized edgeless simple cubic lattice structures.

**Figure 19 materials-16-02870-f019:**
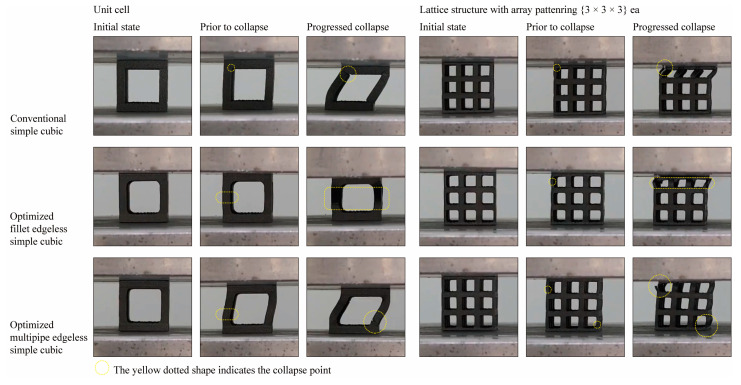
Compressive deformation response of optimized edgeless simple cubic lattice structures.

**Figure 20 materials-16-02870-f020:**
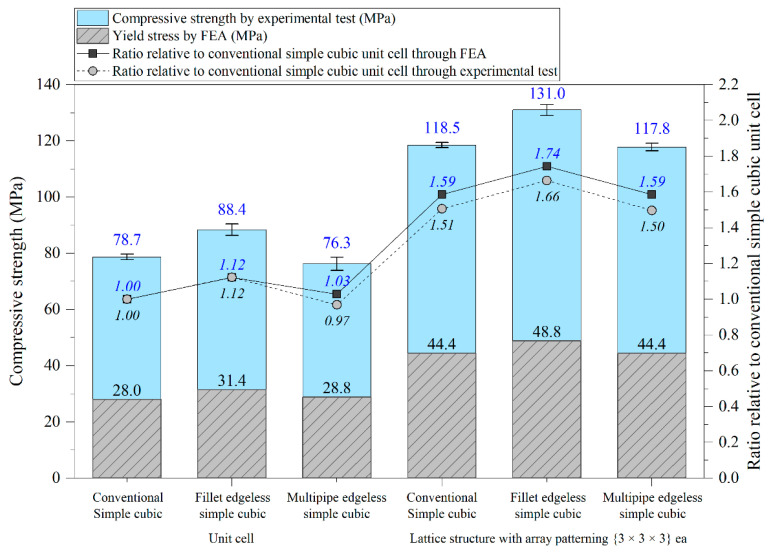
Compressive strength of optimized edgeless simple cubic lattice structures.

**Table 1 materials-16-02870-t001:** Design database for fillet edgeless simple cubic unit cell with relative densities of 0.1, 0.2, and 0.3.

Relative Density = 0.1			Relative Density = 0.2			Relative Density = 0.3		
r/s *^1^	$\overset{-}{r}/s$ *^2^	$\overset{-}{r}/r$ *^3^	r/s	$\overset{-}{r}/s$	$\overset{-}{r}/r$	r/s	$\overset{-}{r}/s$	$\overset{-}{r}/r$
0.1106	-	-	0.1623	-	-	0.2056	-	-
0.1106	0.01	0.0904	0.1623	0.01	0.0616	0.2055	0.01	0.0487
0.1105	0.02	0.1810	0.1622	0.02	0.1233	0.2054	0.02	0.0974
0.1104	0.03	0.2717	0.1621	0.03	0.1851	0.2053	0.03	0.1461
0.1103	0.04	0.3626	0.1620	0.04	0.2469	0.2052	0.04	0.1949
0.1102	0.05	0.4537	0.1618	0.05	0.3090	0.2050	0.05	0.2439
0.1100	0.06	0.5455	0.1616	0.06	0.3713	0.2047	0.06	0.2931
0.1096	0.07	0.6387	0.1613	0.07	0.4340	0.2044	0.07	0.3425
0.1094	0.08	0.7313	0.1609	0.08	0.4972	0.2041	0.08	0.3920
0.1091	0.09	0.8249	0.1606	0.09	0.5604	0.2036	0.09	0.4420
0.1087	0.10	0.9200	0.1602	0.10	0.6242	0.2031	0.10	0.4924
0.1082	0.11	1.0166	0.1597	0.11	0.6888	0.2026	0.11	0.5429
0.1080	0.12	1.1111	0.1593	0.12	0.7533	0.2020	0.12	0.5941
0.1074	0.13	1.2104	0.1587	0.13	0.8192	0.2015	0.13	0.6452
0.1069	0.14	1.3096	0.1582	0.14	0.8850	0.2009	0.14	0.6969
0.1064	0.15	1.4098	0.1576	0.15	0.9518	0.2002	0.15	0.7493
0.1059	0.16	1.5109	0.1570	0.16	1.0191	0.1955	0.16	0.8020
0.1052	0.17	1.6160	0.1563	0.17	1.0877	0.1987	0.17	0.8556
0.1047	0.18	1.7192	0.1556	0.18	1.1568	0.1980	0.18	0.9091
0.1038	0.19	1.8304	0.1548	0.19	1.2274	0.1971	0.19	0.9640
0.1031	0.20	1.9399	0.1540	0.20	1.2987	0.1962	0.20	1.0194
N/A *^4^	N/A	N/A	0.1531	0.21	1.3717	0.1953	0.21	1.0753
N/A	N/A	N/A	0.1523	0.22	1.4445	0.1943	0.22	1.1323
N/A	N/A	N/A	0.1515	0.23	1.5182	0.1933	0.23	1.1899
N/A	N/A	N/A	0.1504	0.24	1.5957	0.1923	0.24	1.2480
N/A	N/A	N/A	0.1495	0.25	1.6722	0.1912	0.25	1.3075

*^1^ r/s: ratio of the strut radius to the edge length. *^2^ $\overset{-}{r}/s$: ratio of the strut radius to the edge length. *^3^ $\overset{-}{r}/r$: ratio of the fillet radius to the strut radius. *^4^ N/A: Not able to design.

**Table 2 materials-16-02870-t002:** FEA results for the fillet edgeless simple cubic unit cells.

Design Parameters for Fillet Edgeless Simple Cubic with a Relative Density of 0.2			Results of FEA			
r/s *^1^	$\overset{-}{r}/s$ *^2^	$\overset{-}{r}/r$ *^3^	Yield Stress of Unit Cell (MPa)	Stress Concentration		
				Total Number of Meshes (ea)	Number of Meshes in the Top 10% of Stress Range (ea)	Percentage of the Number of Meses in the Top 10% of the Stress Range (%)
0.1623	-	-	28.0	521,249	5032	0.97
0.1622	0.02	0.1233	27.7	524,870	5010	0.95
0.1620	0.04	0.2469	27.8	529,862	5206	0.98
0.1616	0.06	0.3713	28.9	522,764	7027	1.34
0.1609	0.08	0.4972	30.0	528,825	9867	1.87
0.1602	0.10	0.6242	30.5	543,516	11,003	2.02
0.1593	0.12	0.7533	31.4	538,276	12,731	2.37
0.1582	0.14	0.8850	31.3	537,120	13,300	2.48
0.1570	0.16	1.0191	31.2	524,071	13,218	2.52
0.1556	0.18	1.1568	31.0	523,350	13,681	2.61
0.1540	0.20	1.2987	30.8	521,775	13,831	2.65
0.1523	0.22	1.4445	30.5	531,064	15,180	2.86
0.1504	0.24	1.5957	30.3	531,067	16,031	3.02

^*1^ r/s: ratio of strut radius to edge length. *^2^ $\overset{-}{r}/s$: ratio of strut radius to edge length. *^3^ $\overset{-}{r}/r$: ratio of fillet radius to strut radius.

**Table 3 materials-16-02870-t003:** FEA results for the multipipe edgeless simple cubic unit cells.

Design Parameters for Fillet Edgeless Simple Cubic with a Relative Density of 0.2			Results of FEA			
r/s *^1^	nodesize/s *^2^	endoffset	Yield Stress of Unit Cell (MPa)	Stress Concentration		
				Total Number of Meshes (ea)	Number of Meshes in the Top 10% of Stress Range (ea)	Percentage of the Number of Meses in the Top 10% of the Stress Range (%)
0.1623	-	-	28.0	521,249	5032	0.97
-	0.1634	0.001	26.1	530,116	3613	0.68
-	0.1630	0.110	26.9	537,315	4831	0.90
-	0.1620	0.350	28.1	531,025	5795	1.09
-	0.1610	0.540	28.7	522,250	6970	1.33
-	0.1600	0.710	28.8	523,742	7365	1.41
-	0.1590	0.865	28.8	516,855	7189	1.39
-	0.1580	1.010	28.6	520,981	7570	1.45
-	0.1462	1.220	27.8	513,029	9995	1.95
-	0.1019	3.850	23.1	479,168	8314	1.74

^*1^ r/s: ratio of the strut radius to the edge length. *^2^ nodesize/s: ratio of the node size to the edge length.

**Table 4 materials-16-02870-t004:** Optimized edgeless simple cubic lattice structures with array pattering {3 × 3 × 3} ea.

Types	Conditions	Unit Cell with Dimensions of 20 mm × 20 mm × 20 mm		Lattice Structure Dimensions of 20 mm × 20 mm × 20 mm with Array Patterning {3 × 3 × 3} ea	
		Design	FEA	Design	FEA
Conventional simple cubic	Morphology	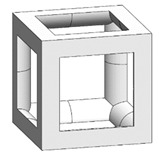	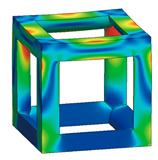	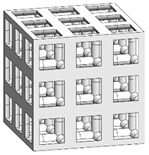	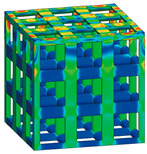
	Design parameter	r/s *^1^ = 0.1623			
	Yield stress (MPa)	28.0		44.4	
Optimized fillet edgeless simple cubic	Morphology	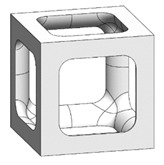	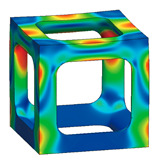	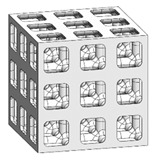	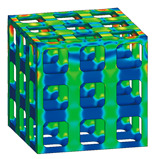
	Design parameter	r/s $=0.1593,\;\overset{-}{r}/s\;$${*}^{2}=0.12,\;\overset{-}{r}/r$ *^3^ = 0.7533			
	Yield stress (MPa)	31.4		48.8	
Optimized Multipipe edgeless simple cubic	Morphology	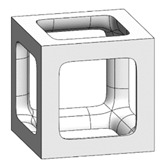	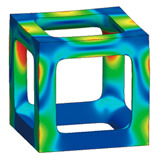	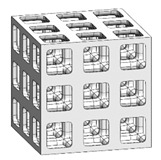	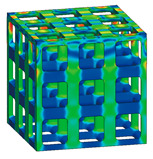
	Design parameter	nodesize/s *^4^ = 0.1600, *endoffset* = 0.710			
	Yield stress (MPa)	28.8		44.4	

*^1^ r/s: ratio of strut radius to edge length. *^2^ $\overset{-}{r}/s$: ratio of strut radius to edge length. *^3^ $\overset{-}{r}/r$: ratio of fillet radius to strut radius. *^4^ nodesize/s: ratio of node size to edge length.

**Table 5 materials-16-02870-t005:** Comparison of FEA and experimental results.

Type	Morphology	FEA		Experimental Test	
		Yield Stress by FEA (MPa)	Ratio Relative to Conventional Simple Cubic Unit Cell	Compressive Strength by Experimental Test (MPa)	Ratio Relative to Conventional Simple Cubic Unit Cell
Unit cell	Conventional Simple cubic	28.0	1.00	78.7	1.00
	Fillet edgeless simple cubic	31.4	1.12	88.4	1.12
	Multipipe edgeless simple cubic	28.8	1.03	76.3	0.97
Lattice structure	Conventional Simple cubic	44.4	1.59	118.5	1.51
	Fillet edgeless simple cubic	48.8	1.74	131.0	1.66
	Multipipe edgeless simple cubic	44.4	1.59	117.8	1.50

## Data Availability

The data presented in this study are available upon request from the corresponding author.

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
