# Peer review of "Design Optimization of Additive Manufactured Edgeless Simple Cubic Lattice Structures under Compression"

_materials, 2023, doi:10.3390/ma16072870_

Round 1
Reviewer 1 Report
1. The actual industrial application or application background of the structure needs a supplementary explanation.
2. The connection between research and the background of additive manufacturing needs more explanation.
3. When a single structure is expanded into multiple, it seems that it is just a simple arrangement increase (see Fig. 6). Considering the actual industrial applications, the author needs to explain whether such a structural design can be applied in practical applications.
4. It is recommended to add verification tests.
5. English expression should be improved.
Author Response
Thank you for your kind review. In this study, methodologies and frameworks were developed for the design optimization of edgeless lattice structures under compression, by introducing fillet and multipipe functions. We will continue to conduct lattice structure research and report the results to the journal. We have revised the following to make this study suitable for MDPI. We will respond faithfully to each comment.

Reviewer 2 Report
The authors did a good simulation, even though the following updates are required before proceeding further.
1. The authors need to explain how the parameters are selected and why the unit cell cube edge (?) length is fixed at 20 mm. Include proper references.
2. How is it possible to reduce the stress concentration by 144%? Explain with proper references.
3. The authors did a good simulation, but it cannot be validated without experimental data.
4. Experimental results and comparisons should be included to understand the authenticity of this simulation.
5. Proper references are lagging throughout the article.
6. It is not able to accept the simulated results for yield strength and stress reduction without confirmation.
Author Response

(The authors gave the same response as above.)

Reviewer 3 Report
This work aims to design edgeless lattice structures with fillet and multipipe functions for reducing the stress concentration of the structures. The effect of edgeless parameters on the yield stress of the lattice structures under compression was investigated. Although the topic is meaningful to the readers, there are some flaws. The presented manuscript should be revised before considering for publishing.
(1) The title of this work is Design Optimization of Additive Manufactured Edgeless Simple Cubic Lattice Structures under Compression. However, there are no descriptions on additive manufactured lattice structures.
(2) What are the edgeless parameters of edgeless lattice structures with fillet function? How to define?
(3) The strength of the edgeless lattice structures should be evaluated by nonlinear FEM instead of linear FEM.
(4) In fig.15, why the yield strength arrive the peak value when the ratio of fillet radius to the length edge is 0.12?
(5) The optimized results should be validated by the experiments.
Author Response

(The authors gave the same response as above.)

Reviewer 4 Report
The paper presents an interesting study of design optimization of edgeless cubic lattice structures under compression loading made by of additive manufacturing. However, the study focuses on a numerical analysis rather than physically testing of the lattice structures to validate the numerical simulations. The authors should provide a justification for not validating the simulations. In addition, the following comments should be addressed:
· Please thoroughly proof-read the manuscript as there are few formatting/grammatical errors.
· Consider revising the title to show that only numerical analysis has been undertaken.
· Sine the focus of the work is on compression, therefore, it should be mentioned in the abstract as well.
· Discussion around manufacturing time and material usage should be provided for the lattice structures.
Author Response

(The authors gave the same response as above.)

Reviewer 5 Report
I think the work is presented in a very effective and clear way. The study is carried out in a very detailed way.
On the other hand, I think it does not contribute in a significant way. Most of the results are consistent with what are well known facts about stress concentrators (roundness is better than sharp edges, etc.).
From the point of view of additive manufacturing, the degree to which these results might be relevant is unclear. Typically, AM lattices have cell sizes and struts of just few millimeters. The level of accuracy described in this manuscript is hardly achievable using AM.
Author Response

(The authors gave the same response as above.)

Round 2
Reviewer 2 Report
Authors incorporate all the suggested corrections.
Author Response
I would like to express my gratitude for your kind review of our manuscript. Your insightful comments and suggestions have greatly contributed to the improvement of our study.
As you may know, in this study, we developed methodologies and frameworks for optimizing the design of edgeless lattice structures under compression, by introducing fillet and multipipe functions. We are pleased to inform you that we will continue to conduct further research on lattice structures and report our findings to the journal.
Once again, thank you for your thoughtful suggestions and insights, which have significantly benefited our manuscript. We greatly appreciate your support and look forward to collaborating with you to bring this manuscript closer to publication in Materials.

Reviewer 3 Report
The authors have responded all the reviewer's comments. The revised manuscript can be published.
Author Response
I would like to express my gratitude for your kind review of our manuscript. Your insightful comments and suggestions have greatly contributed to the improvement of our study.
As you may know, in this study, we developed methodologies and frameworks for optimizing the design of edgeless lattice structures under compression, by introducing fillet and multipipe functions. We are pleased to inform you that we will continue to conduct further research on lattice structures and report our findings to the journal.
Once again, thank you for your thoughtful suggestions and insights, which have significantly benefited our manuscript. We greatly appreciate your support and look forward to collaborating with you to bring this manuscript closer to publication in Materials.
Sincerely,

Reviewer 4 Report
The authors have responded to most of my comments. However, the experimental validation has raised more questions that have not been addressed. There are large differences between the yield stress from FEA and compressive strength from experimentation. The reason for validating simulations is to ensure that they are representing what would happen in reality. Please explain the reasoning for such large differences as this would mean the simulations have not been validated and are not representing reality. The authors should discuss the error percentage and whether it is acceptable for such a product/test.
Author Response

(The authors gave the same response as above.)

Reviewer 5 Report
The addition of the experiments to back up the theoretical findings have improved significantly the work.
However, some additional work might be necessary. Please consider the following comments:
Abstract: It must include a mention of experiments and the validation of the FE model.
Line 35: "lattice structures have biocompatibility..." is not an accurate description; biocompatibility depends on the base material and not the geometry. However, lattice structures are a good fit for biomedical implants because of their porosity and integration potential to bone structures.
Line 55: PBF is only one of many AM methods that can be used to produce lattices. ME and VP can also be used for that purpose.
Section 2.5: How many specimens were tested for each geometry? Just one? Given the variability in AM processes, it is desirable that multiple tests are performed for each type of specimen when dealing with AM lattices. Consider the standard ISO 13314.
Line 256-259: Please rewrite the sentence. It is confusing. Maybe change the order of sentences 1 and 2 may improve clarity.
Line 272: is upper loading area the same as the original cross-section area?
Fig. 18: Was any treatment applied to the specimens after the AM process? If so, it should be included in Section 2.5
Line 405-413: Analysis should focus on the stress concentrators in FE model vs where is the damage in the specimens, just like in previous lines. Because the FE model is limited to the elastic region, it is hard to correlate plastic deformation and FE results.
Lines 420-424: There's no need to repeat the values; they are already shown in the table.
Table 5: Although the FE model reproduce qualitatively the ratios between conventional and optimized unit cells, there is still a significant difference between experimental and numerical values. This difference must be explained and point out probable causes for this.
Conclusions: It must include a brief discussion about the experimental results and the validation of the model.
Author Response

(The authors gave the same response as above.)

Round 3
Reviewer 4 Report
I am afraid without validating the simulations results, their value is diminished. There are two ways of going about this. The authors can remove the experimental discussion and go with a simulation study alone highlighting their design optimisation and stating that due to the design complexity, validation model needs to be developed or something along these lines. The other option is to develop a validation model with an acceptable error percentage.
Author Response
I would like to express my sincere gratitude for your thoughtful review of our manuscript. Your insightful comments and suggestions have significantly contributed to the improvement of our study.
I fully understand and appreciate your concerns regarding our research. However, the purpose of our study is to present an optimization methodology for designing edgeless lattice structures. We introduced a fillet and multipipe design function to smooth the corners of a simple cubic lattice, and provided data for optimal edgeless design. We believe that our data will serve as a valuable reference for researchers studying unit cells and lattice structures in terms of design.
Moreover, we used FEA to verify the edgeless lattice structure by reviewing the yield stress and stress concentration using nonlinear static analysis. Our results showed that the yield stress and stress concentration were significantly improved by the introduction of edgeless.
In addition to your review comments, other reviewers recommended manufacturing test specimens using AM technology and measuring the compressive strength. Although our research funding has ended, we manufactured test specimens and conducted experimental tests at a cost of approximately 2000 USD. The maximum compressive strength was measured, which showed that the compressive strength of lattice structure improved by the introduction of edgeless.
Our study has three main results: (1) proposal of an edgeless design method for lattice structures, (2) validation through FEA showing improvement of the yield stress and stress concentration of edgeless lattice structures, and (3) validation through experimental tests showing improvement of the maximum compressive strength of edgeless lattice structures.
However, as you pointed out, there is a difference between the results of FEA and experimental tests. The FEA was interpreted based on the yield strength of the mesh in simulation, while the compressive strength test was measured based on the maximum compressive strength at the time the lattice structure collapsed. Therefore, there is a difference in the result value.
The purpose of the experimental test is to verify the improvement in the maximum compressive strength of the edgeless lattice structure.
To address this issue, we believe that it is necessary to develop a process that interprets fracture mode as a standard from the FEA. From the point of the experimental test, it is necessary to observe the detailed stress-strain curve and observe the curve changes in the elastic region and plastic region. Based on this, we believe that the relationship between the FEA results and the verification test results of the edgeless lattice structure can be derived more clearly. As our research team continues to conduct research in this field, we will review these uncertainties and continuously report the research results in the future.
We are well aware of the limitations of our study. We will revise the conclusion section of the manuscript to eliminate any misunderstandings for the readers and present valid results clearly.
Once again, we appreciate your kind review and suggestions, and kindly request your positive consideration.
